# Lack of Sex Disparity in Oral Anticoagulation in Atrial Fibrillation Patients Presenting with Ischemic Stroke in a Rural Population

**DOI:** 10.3390/jcm10204670

**Published:** 2021-10-12

**Authors:** Eric Koza, Johan Diaz, Durgesh Chaudhary, Shima Shahjouei, Jiang Li, Vida Abedi, Ramin Zand

**Affiliations:** 1Geisinger Commonwealth School of Medicine, Scranton, PA 18510, USA; ekoza@som.geisinger.edu (E.K.); jdiaz01@som.geisinger.edu (J.D.); 2Department of Neurology, Neuroscience Institute, Geisinger Health System, 100 North Academy Ave, Danville, PA 17822, USA; dpchaudhary@geisinger.edu (D.C.); sshahjouei@geisinger.edu (S.S.); 3Department of Molecular and Functional Genomics, Geisinger Health System, Danville, PA 17822, USA; jli@geisinger.edu (J.L.); vabedi@geisinger.edu (V.A.); 4Biocomplexity Institute, Virginia Tech, Blacksburg, VA 24061, USA

**Keywords:** ischemic stroke, atrial fibrillation, oral anticoagulants, undertreatment, sex disparity, CHA_2_DS_2_-VASc

## Abstract

Various studies on oral anticoagulants (OAC) use among atrial fibrillation (AF) patients have shown high rates of undertreatment and the presence of sex disparity. This study used the ‘Geisinger Neuroscience Ischemic Stroke’ (GNSIS) database to examine sex differences in OAC treatment among ischemic stroke patients with the pre-event diagnosis of AF in rural Pennsylvania between 2004 and 2019. We examined sex disparities in OAC undertreatment and associated risks based on age group and ischemic stroke year. A total of 1062 patients were included in the study and 1015 patients (96%) had CHA_2_DS_2_-VASc score ≥ 2, of which 549 (54.1%) were women. Undertreatment rates were not statistically significant between men and women in the overall cohort (50.0% vs. 54.5%, *p* = 0.18), and male sex was not found to be a significant factor in undertreatment (OR 0.82, 95% CI 0.62–1.09, *p* = 0.17). The result persisted even when patients were divided into four age groups and two groups based on the study time period. The undertreatment rates in both sex groups remained consistent following the introduction of novel oral anticoagulants. In conclusion, there was no evidence of sex disparity with respect to OAC treatment, even after stratifying the cohort by age and ischemic stroke year.

## 1. Introduction

Atrial fibrillation (AF) is a major risk factor for stroke incidence and the most significant cardiac arrhythmia worldwide [1,2]. It is estimated that around 7.6 million Americans have suffered a stroke, with women carrying a higher lifetime risk when compared to men [2]. The increased risk in women has been translated to the clinical practice through the inclusion of female sex within risk stratification models for stroke management, such as the CHA_2_DS_2_-VASc score [3,4,5,6]. Current guidelines by the American Heart Association/American College of Cardiology/Heart Rhythm Society (AHA/ACC/HRS) recommend oral anticoagulation (OAC) therapy for all AF patients with a CHA_2_DS_2_-VASc ≥ 2, and those with a CHA_2_DS_2_-VASc of 1 should be considered for treatment with OAC or antiplatelets [3].

Despite the AHA/ACC/HRS guidelines for the management of patients with atrial fibrillation [3], some studies have suggested that women have a higher risk of OAC undertreatment than men [7,8,9,10], while others have reported no significant differences [11,12]. A previous study using a national registry in the United States found that women had a higher risk of undertreatment across all CHA_2_DS_2_-VASc scores [9], while an investigation in Europe found that a higher proportion of female patients received OAC treatment in contrast to males [13]. Further, a recent international study found no differences in anticoagulation use between women and men globally except for a sex disparity specific to North America [11]. While the authors attributed these results to differences in guideline recommendations [11], there is still uncertainty about sex inequalities in OAC treatment due to the combined contradictory findings.

In a previous study, we investigated the prevalence and factors associated with AF undertreatment in patients with stroke outcomes [14]. The current analysis aimed to determine whether sex influenced AF treatment among a rural population of stroke patients in central and northeast Pennsylvania, USA. The study examined sex disparities by evaluating undertreatment rates and risk associations, stratifying patients based on age groups and index stroke dates, mainly considering the introduction of non-vitamin K antagonist oral anticoagulants (NOACs) in 2010. 

## 2. Materials and Methods

### 2.1. Data Source and Study Population 

The study used a retrospective cohort analyzing data from the “Geisinger Neuroscience Ischemic Stroke (GNSIS)” registry, a database of ischemic stroke patients at Geisinger which includes demographics, family history, and clinical and past medical history. Geisinger is an integrated system delivering healthcare in rural Pennsylvania to approximately 2.6 million people throughout 43 counties. Patients were included in GNSIS if they had a primary diagnosis of ischemic stroke based on the International Classification of Diseases, Ninth/Tenth Revision, Clinical Modification (ICD-9-CM or ICD-10-CM) during a hospital encounter of at least 24 h, and a magnetic resonance imaging of the brain in the same encounter. Further details on the data extraction and pre-processing for the GNSIS database are provided in previously published study articles [14,15,16]. The Geisinger Institutional Review Board reviewed and approved this study as a “non-human subject research” for using de-identified information.

### 2.2. Evaluation of Sex Disparities 

The study included ischemic stroke patients ≥ 18 years old with an AF diagnosis ICD (ICD-9-CM and ICD-10-CM) code at any time before the stroke index date, and it excluded patients with an AF diagnosis on the index stroke date. The study analyzed point prevalence among men and women, and patients’ distribution according to CHA_2_DS_2_-VASc score. Sex was studied as a risk factor for undertreatment in patients with CHA_2_DS_2_-VASc ≥ 2 stratified based on age group or NOACs use. Alternatively, CHADS_2_ was employed for risk stratification with a score of ≥2 denoting patients at high risk of stroke.

To study the impact of NOACs on sex disparity in undertreatment, patients were stratified based on their index stroke date. The first group contained patients with an index stroke date between 2004 and 2010, while the second group included patients with an index stroke date between 2011 and 2019. To examine the sex disparity in OAC treatment in different age groups, the patients were also divided into four subgroups based on age, and undertreatment was examined in each subgroup.

The HAS-BLED score is used by clinicians to evaluate the risk of bleeding in AF patients and assesses the history of uncontrolled hypertension, renal or liver disease, stroke, bleeding, labile INR, age ≥ 65 years, medications, and alcohol use [3]. For this study, a limited HAS-BLED score was calculated from the liver and renal function, age, medications predisposing to bleeding, and history of stroke, TIA, and bleeding diagnoses. The differences in the limited HAS-BLED scores between the male and female patients were examined along with its association with sex disparity in undertreatment.

### 2.3. Statistical Analysis

In the study, demographic and comorbidity characteristics for the population were summarized using descriptive statistics. Continuous variables were presented as mean ± standard deviation or median with interquartile range (IQR), and categorical variables were presented as counts and percentages. Statistical analysis among groups included Pearson’s chi-squared test or Fisher’s exact test for the categorical variables, and analysis of variance (ANOVA) or the Kruskal–Wallis test for continuous variables. Multiple logistic regression was performed to examine the association of gender with undertreatment in AF patients, adjusting for age and comorbidities, while the goodness of fit was evaluated using the Hosmer–Lemeshow test. Variables with missingness, such as the National Institutes of Health Stroke Scale (NIHSS), were not included in regression analysis. The alpha value for all *p*-values was set to 0.05. R version 4.0.3 (R Foundation for Statistical Computing, Vienna, Austria) was used for all statistical analyses.

## 3. Results

### 3.1. Patient Characteristics of Study Population

Evaluation of ischemic stroke patients from the GNSIS database yielded 1062 patients with an AF diagnosis before the index stroke date (Figure 1). Of these patients, 506 (47.6%) were men, and 556 (52.4%) were women (Table 1). The median age at AF diagnosis was 72.3 years (IQR 64.4–79.2) for men and 78.9 years (IQR 69.8–84.5) for women (*p* < 0.001). The median age at the index date of stroke was 76.9 years (IQR 68.1–82.9) for men and 82.6 years (IQR 74.9–88.3) for women (*p* < 0.001). Additional patient characteristics are summarized in Table 1. In terms of medication, there were no significant differences between men and women using antiplatelets only (116 (22.9%) vs. 115 (20.7%); *p* = 0.418), anticoagulants only (129 (25.5%) vs. 146 (26.3%); *p* = 0.830), or anticoagulants and antiplatelets (120 (23.7%) vs. 106 (19.1%); *p* = 0.076). There was a significant difference between the CHA2DS_2_-VASc median score (male median, 4; IQR, 3–5 vs. female median, 5; IQR, 4–6; *p* < 0.001). Further comparison of CHA2DS_2_-VASc scores distribution based on sex can be seen in Figure 2. 

### 3.2. Undertreatment of Atrial Fibrillation

After dividing the distribution of CHA2DS_2_-VASc into three categories, 0, 1, and 2+ (Table 1), 1015 (95.6%) patients were included in the 2+ category. Of the 1015 patients, 466 (45.9%) were men, and 549 (54.1%) were women (Figure 1). Further evaluation of these patients showed that based on current treatment guidelines, 233 (50.0%) men were not receiving adequate therapy, while 299 (54.5%) women were also not receiving adequate therapy (*p* = 0.18), as seen in Table 2. Overall, the male sex was not found to have a statistically significant association with undertreatment in multiple logistic regression (OR 0.82, 95% CI 0.62–1.09, *p* = 0.17) (Figure 3). 

### 3.3. Anticoagulant Undertreatment Rate Based for Different Age Groups

To examine sex differences in atrial fibrillation, patients were first stratified into four groups based on age quartiles (Table 2). Group 1 consisted of 254 patients aged 44.4–72.8 years, Group 2 had 254 patients aged 72.8–80.6 years, Group 3 included 254 patients aged 80.6–86.6 years, and Group 4 with 253 patients older than 86.6 years. The OAC undertreatment rate was not significantly different in any of the four age groups (Table 2), nor was male sex a significant predictor in the multiple logistic regression on undertreatment (Figure 3). 

### 3.4. Anticoagulant Undertreatment Rate Based on Index Stroke Year

When grouping the patients based on the index stroke year, the first group (2004–2010) contained 221 patients with 112 (50.7%) identified as undertreated with anticoagulants, 58 (51.3%) men, and 54 (50.0%) women (Table 2). Undertreatment rates between the sexes did not show a statistically significant difference (*p* = 0.95). Additionally, sex was not associated with undertreatment rate (OR 1.46, 95% CI 0.75–2.87, *p* = 0.26) in Figure 3. In the second group (2011–2019), there were 794 patients, of which 420 (52.9%) were undertreated with anticoagulants, 175 (49.6%) men and 245 (55.6%) women (Table 2). No statistically significant difference was observed between the undertreatment of men and women (*p* = 0.11) in Table 2, nor with the association of male sex and undertreatment in multiple logistic regression (OR 0.73, CI 0.53–1.01, *p* = 0.06) in Figure 3. Overall, undertreatment rates following the introduction of NOACs were comparable to previous years without significant differences in OAC usage between sex groups. 

## 4. Discussion

The results from our rural AF patient population with ischemic stroke outcomes indicate the presence of undertreatment according to guideline-recommended OAC of about 50% in men and 55% in women with a CHA_2_DS_2_-VASc ≥ 2; however, sex was not shown to be a statistically significant risk factor for OAC undertreatment. When dividing the population based on age and index stroke year, both analyses revealed no apparent sex disparity and no risk association between sex and undertreatment. Following the introduction of NOACs in 2010, the undertreatment rates remained consistent between the 2004–2010 and 2010–2019 groups, without significant sex disparities or associated risk. Similarly, risk stratification using CHADS_2_ scores did not show a sex disparity or risk association with undertreatment (Appendix A, Appendix A). Several associated factors were more prevalent in men, such as dyslipidemia, diabetes, peripheral vascular disease, and a history of myocardial infarction. Women were diagnosed with AF at an older age, had an older age at index stroke date, and a higher median baseline CHA2DS2-VASc, which aligned with populations in similar studies [5,9,10,12,13,17,18].

In contrast, recent studies have identified potential sex disparities in OAC undertreatment (Appendix A). Retrospectives studies have found lower odds of OAC initiation and an increased likelihood of undertreatment among high-risk female patients with Medicare and commercial insurance [7,8]. Similarly, a study using the PINNACLE National Cardiovascular Data Registry found that women had a higher risk of undertreatment across all levels of the CHA_2_DS_2_-VASc scores [9], while a study from the Euro Observational Research Programme Pilot survey on atrial fibrillation (EORP-AF) reported that more women with CHA_2_ DS_2_-VASc ≥ 2 received OAC than men [13]. However, other researchers have found comparable results to those presented in this study (Appendix A). A prospective cohort study in China reported no sex difference in OAC treatment among AF patients with CHA_2_DS_2_-VASc ≥ 2, but women represented a smaller percentage of the patient population [12]. Two international studies found no significant sex disparities in OAC therapy worldwide [11,19]. In one of these studies, the authors described a sex disparity specific to North America that was attributed to differences in thresholds for OAC initiation due to varying guideline recommendations [11]. 

Our statistical analysis and logistic regression models of undertreatment showed no association for male sex with a decreased risk of OAC undertreatment, differing from previous studies (Appendix A). Two investigations in Italy and Sweden stratified AF patients based on age and year cohorts found that women in the ≥75 age group had an increased risk of OAC undertreatment, even with the emergence of NOACs and updated European Society of Cardiology Guidelines [10,20,21,22]. In general, prior studies concur that increasing age seems to correlate to an increased risk of AF undertreatment, especially among patients ≥70 or 80 years old [23,24,25,26,27]. There have been reports of an aging paradox, where patients ≥70 or 80 years old were less likely to receive OAC therapy despite a heightened risk of stroke [23,24,25,26,27]. Our previous study using the GNSIS registry found that less than half of our high-risk patients received OAC treatment according to guidelines [14]. These results were comparable to other studies in the USA [23,24]. However, we did not observe an increased risk of undertreatment associated with age [14].

There were several strengths and limitations present within our study. The EHR data collected provided a wealth of variables for each patient, as well as a wide time frame for evaluation, allowing a more robust analysis of differences in patient characteristics between sexes. While we had a large cohort comprised of patients from multiple study centers, our cohort lacked racial and socioeconomic diversity seen in previous related studies [7,9,10,17], in addition to the patient population being restricted to only rural areas, thus possibly limiting the external validity of our results. 

Another limitation may be the use of CHA_2_ DS_2_-VASc to assess stroke risk and undertreatment rates in AF patients from 2004 to 2019. In the United States, CHA_2_ DS_2_-VASc was implemented in 2014, and prior to this guideline update, CHADS_2_ was used for risk stratification [3]. However, in our previous study, we found similar undertreatment rates using CHA_2_ DS_2_-VASc and CHADS_2_ scores [14], and our results in this study are consistent even when using CHADS_2_ for risk stratification. Other limitations include the lack of a complete HAS-BLED score (Appendix A), used by clinicians to identify patients at risk for bleeding [3,6,28] and to potentially examine sex disparities [7,8,9,11,13,19]. 

Lastly, the use of ICD-9-CM and ICD-10-CM posed potential constraints. AF diagnosis was measured using ICD codes without electrocardiogram (EKG) confirmation. Furthermore, ICD codes do not provide detailed clinical information and sex-related preferences. Previous studies have found that women tend to present with more symptoms, older age, and receive more conservative treatments [12,13]. Additionally, ICD codes fail to give insight into other possible external factors affecting treatment, such as specific patient and physician preferences, and the decision process in treatment choice, as well as accurate insurance information. While medication data in EHR can provide information on whether a patient was prescribed a certain medication, determining the patient compliance remains a challenge [29]. 

## 5. Conclusions

This study found that OAC treatment for AF in ischemic stroke patients remains low according to guidelines in both men and women; however, there does not appear to be a sex disparity, even when stratifying patients by age and index ischemic stroke date. Sex was not associated with OAC undertreatment in our rural population. 

## Figures and Tables

**Figure 1 jcm-10-04670-f001:**
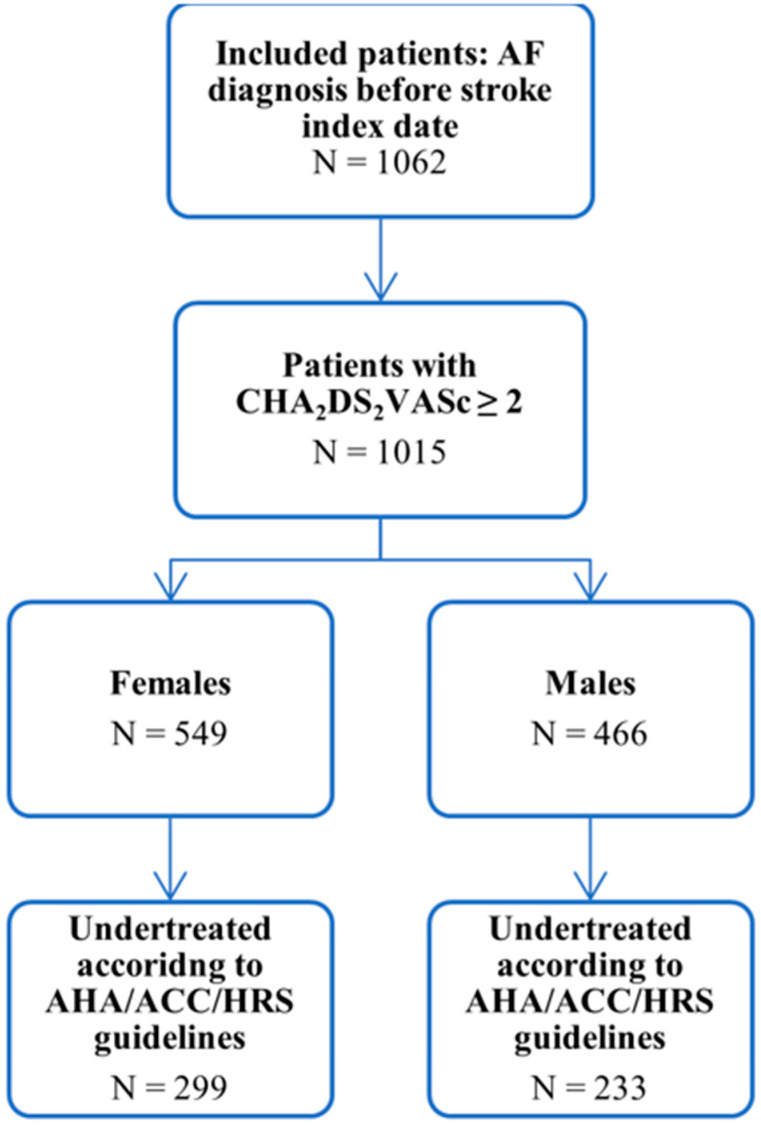
Flowchart of patients with atrial fibrillation in the study.

**Figure 2 jcm-10-04670-f002:**
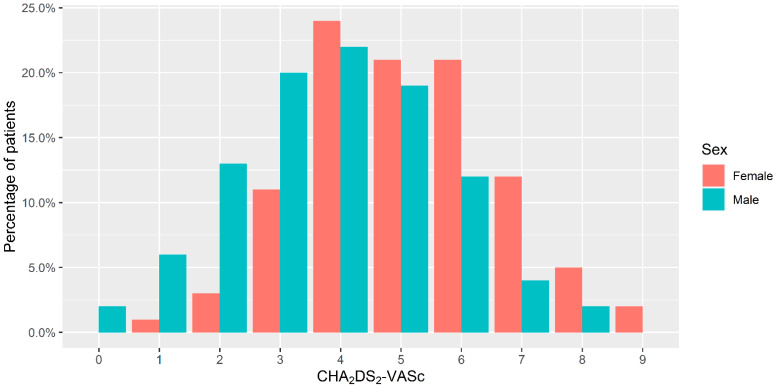
CHA_2_DS_2_-VASc score patient distribution based on sex.

**Figure 3 jcm-10-04670-f003:**
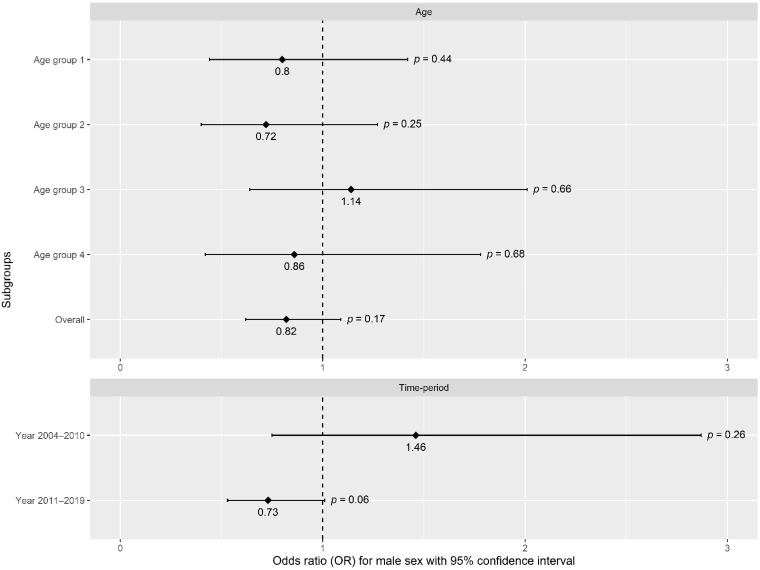
Logistic regression for patients with a CHA_2_DS_2_-VASc score ≥ 2 stratified by age groups and index stroke date.

**Table 1 jcm-10-04670-t001:** Male and female ischemic stroke patient characteristics with a diagnosis of atrial fibrillation before index stroke event.

Variable	Overall	Female	Male	*p*-Value
**Number of patients**	1062	556	506	
**Age at atrial fibrillation diagnosis in years, median (IQR)**	75.5 (67.3, 82.5)	78.9 (69.8, 84.5)	72.3 (64.4, 79.2)	<0.001 *
**Age at index stroke event in years, median (IQR)**	80.0 (71.5, 86.3)	82.6 (74.9, 88.3)	76.9 (68.1, 82.9)	<0.001 *
**CHA_2_DS_2_-VASc at baseline, median (IQR)**	4.0 (3.0, 6.0)	5.0 (4.0, 6.0)	4.0 (3.0, 5.0)	<0.001 *
**CHA_2_DS_2_-VASc**				<0.001 *
**0**	10 (0.9)	0 (0.0)	10 (2.0)	
**1**	37 (3.5)	7 (1.3)	30 (5.9)	
**2+**	1015 (95.6)	549 (98.7)	466 (92.1)	
**Dyslipidemia, *n* (%)**	740 (69.7)	368 (66.2)	372 (73.5)	0.011 *
**Heart failure, *n* (%)**	397 (37.4)	209 (37.6)	188 (37.2)	0.934
**Hypertension, *n* (%)**	900 (84.7)	472 (84.9)	428 (84.6)	0.957
**Diabetes, *n* (%)**	392 (36.9)	188 (33.8)	204 (40.3)	0.033 *
**Past ischemic stroke, *n* (%)**	115 (10.8)	66 (11.9)	49 (9.7)	0.295
**Transient ischemic attack, *n* (%)**	135 (12.7)	74 (13.3)	61 (12.1)	0.603
**Other thromboembolism, *n* (%)**	79 (7.4)	46 (8.3)	33 (6.5)	0.332
**Myocardial infarction, *n* (%)**	215 (20.2)	84 (15.1)	131 (25.9)	<0.001 *
**Peripheral vascular disease, *n* (%)**	275 (25.9)	119 (21.4)	156 (30.8)	0.001 *
**Hypercoagulative State, *n* (%)**	13 (1.2)	9 (1.6)	4 (0.8)	0.271
**Chronic liver disease, *n* (%)**	42 (4.0)	16 (2.9)	26 (5.1)	0.084
**Cirrhosis, *n* (%)**	12 (1.1)	6 (1.1)	6 (1.2)	1.000
**Chronic kidney disease, *n* (%)**	391 (36.8)	217 (39.0)	174 (34.4)	0.133
**End-stage renal disease ESRD, *n* (%)**	35 (3.3)	13 (2.3)	22 (4.3)	0.097
**Past hemorrhagic stroke, *n* (%)**	37 (3.5)	18 (3.2)	19 (3.8)	0.770
**Medications**				
**Antiplatelets, *n* (%)**	231 (21.8)	115 (20.7)	116 (22.9)	0.418
**Anticoagulants, *n* (%)**	275 (25.9)	146 (26.3)	129 (25.5)	0.830
**Anticoagulant and Antiplatelet, *n* (%)**	226 (21.3)	106 (19.1)	120 (23.7)	0.076
**Statins, *n* (%)**	501 (47.2)	242 (43.5)	259 (51.2)	0.015 *
**Antihypertensives, *n* (%)**	546 (51.4)	272 (48.9)	274 (54.2)	0.101
**Medical insurance type, *n* (%) ^†^**				<0.001 *
**Commercial**	132 (12.8)	55 (10.2)	77 (15.7)	
**Health Maintenance Organization (HMO)**	352 (34.1)	174 (32.2)	178 (36.2)	
**Medicaid**	13 (1.3)	7 (1.3)	6 (1.2)	
**Medicare**	518 (50.2)	297 (55.0)	221 (44.9)	
**Self-Pay**	1 (0.1)	1 (0.2)	0 (0.0)	
**Special Billing**	9 (0.9)	6 (1.1)	3 (0.6)	
**Veterans Affairs (VA)**	7 (0.7)	0 (0.0)	7 (1.4)	
**Smoking status, *n* (%)**				<0.001 *
**Current smoke**	87 (8.2)	25 (4.5)	62 (12.3)	
**Past smoker**	408 (38.4)	140 (25.2)	268 (53.0)	
**Never smoker**	520 (49.0)	368 (66.2)	152 (30.0)	
**Unknown**	47 (4.4)	23 (4.1)	24 (4.7)	
**NIHSS at index stroke event, median (IQR) ^#^**	5.0 (2.0, 9.0)	6.0 (3.0, 12.0)	4.0 (2.0, 6.0)	0.001 *
**All-cause mortality within 1 year of index stroke, *n* (%)**	303 (28.5)	164 (29.5)	139 (27.5)	0.508
**Recorded encounter count per year between diagnosis of atrial fibrillation and index stroke, median (IQR)**	6.3 (3.0, 11.0)	6.9 (3.0, 11.0)	6.0(3.0, 10.8)	0.404
**Time in years between diagnosis of atrial fibrillation and index stroke, median (IQR)**	2.8 (1.0, 5.6)	2.9 (1.0, 5.5)	2.6 (1.0, 5.6)	0.899

* Significant *p*-value; ^†^ Medical insurance data available for 1032 patients (540 female and 492 male patients); ^#^ NIHSS available for only 261 patients (136 female and 125 male patients).

**Table 2 jcm-10-04670-t002:** Anticoagulant undertreatment rates for patients with a CHA_2_DS_2_-VASc score ≥ 2 stratified by age groups and index stroke date.

		All Patients			Undertreated			
		Total	Female	Male	Total	Female	Male	*p*-Value
	**All Groups**	1015	549	466	532 (52.4%)	299 (54.5%)	233 (50.0%)	0.175
**Stratified by Age**	**Group 1 (44.4–72.8 years)**	254	110	144	141 (55.5%)	64 (58.2%)	77 (53.5%)	0.535
	**Group 2 (72.8–80.6 years)**	254	112	142	120 (47.2%)	59 (52.7%)	61 (43.0%)	0.157
	**Group 3 (80.6–86.6 years)**	254	138	116	122 (48.0%)	64 (46.4%)	58 (50.0%)	0.653
	**Group 4 (>86.6 years)**	253	189	64	149 (58.9%)	112 (59.3%)	37 (57.8%)	0.955
**Stratified by Index Date**	**2004–2010**	221	108	113	112 (50.7%)	54 (50.0%)	58 (51.3%)	0.950
	**2011–2019**	794	441	353	420 (52.9%)	245 (55.6%)	175 (49.6%)	0.108

## Data Availability

The data presented in this study are available on request from the corresponding author. The data are not publicly available due to institutional policies requiring data-sharing agreement.

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
