# Peer review of "Lack of Sex Disparity in Oral Anticoagulation in Atrial Fibrillation Patients Presenting with Ischemic Stroke in a Rural Population"

_jcm, 2021, doi:10.3390/jcm10204670_

Round 1

Reviewer 1 Report

In this retrospective study including 1062 patients from the GNSIS database, the authors investigated the potential sex difference in oral anticoagulants (OAC) treatment among ischemic stroke patients with the pre-event diagnosis of atrial fibrillation (AF) in rural Pennsylvania between 2003 and 2019. They showed that OAC undertreatment rates were not statistically different between the men and women and that male sex was not found to be a significant factor in undertreatment, regardless of age groups and study time-period. The main strength of the study is the large sample size and its main weaknesses are the retrospective design and the fact that the manuscript, and especially the results section because of multiple analyses, is not so easy to read. I have additional concerns that need to be discussed. Please consider the following comments. 

General comments

1. The rationale of the study is not sufficiently explained and you should clearly explain at the end of the introduction what your hypothesis is. Why should gender influence the OAC undertreatment rate? Please clarify.

2. Overall, the manuscript is not easy to read and would benefit from simplification and clarification.

Specific comments

3. Why did you only include rural population? This limits the external validity of your results and this point should be discussed in the limitations of the study. Please justify.

4. The Results section is not easy to read. There are too many analyses that dilute your main message. It would be helpful for readers to summarize your key findings with figures and to remove some of the secondary analyses.

5. The discussion is too long and could be shortened by a third without altering your message. Please focus more on your results. In addition, some parts of the discussion should be moved to the introduction to better explain the rationale of the study. Please change introduction and discussion accordingly.

6. Please correct all the spelling errors in the manuscript.

Reviewer 2 Report

No signficant factors showed the relation for disparities in OAC use from the era of VKA to DOAC.  Limited information about the decision process in treatment choice. 

Will insurance or physicians preference affect the treatment choice?

Round 2

Reviewer 1 Report

The authors have taken into account all the comments of the reviewers. I have no further comments. Best regards.

Author Response

We would also like to thank the reviewer for their time and effort to provide us with valuable feedback on this paper. 

Reviewer 2 Report

Better statistical process.

Author Response

We would also like to thank the reviewer for their time and effort to provide us with valuable feedback on this paper. We have made changes to the manuscript to reflect the suggestions that have been provided.  We hope our edits and responses below can address these concerns.

Point 1: Better statistical process.

Response 1: Changes have been made to the methods and statistics for better description according to the suggestions.